# OpenReview forum: "Unmasking Backdoors: An Explainable Defense via Gradient-Attention Anomaly Scoring for Pre-trained Language Models"
_ICLR.cc/2026/Conference — ICLR 2026 Poster_

### Official Review · Reviewer_X2fK · 2025-10-29

**Soundness:** 3
**Presentation:** 3
**Contribution:** 3
**Rating:** 6
**Confidence:** 5

**Summary:**

This paper introduces X-GRAAD, a novel inference-time defense framework for detecting and mitigating backdoor attacks in pre-trained language models (PLMs). The method's core idea is that backdoor trigger tokens, when activated, abnormally dominate the model's attention and gradient attribution signals simultaneously. The proposed method operationalizes this insight by combining these two signals to assess the anomaly score of each token. It identifies malicious inputs by searching for a single "peak" token with an exceptionally high score. If such a token is found, it is neutralized via a noise injection mechanism before the model generates its final prediction. The authors demonstrate experimentally that this method effectively reduces Attack Success Rates (ASR) while maintaining high Clean Accuracy (CACC). The paper also highlights the method's explainability, showing its ability to localize trigger tokens.

**Strengths:**

1. **Novel Core Hypothesis.** The core hypothesis—that backdoor triggers manifest as strong, simultaneous anomalies in both attention and gradient channels—is an intuitive and novel insight. Combining these two distinct attribution modalities for anomaly detection is a strong starting point.
2. **Interpretability.** The method not only provides a defense but also offers interpretability via attribution scores (as shown in Figures 2 and 5), helping to localize and understand the behavior of backdoor triggers, which is a valuable feature.
3. **High Practicality (Efficiency and Simplicity).** As an inference-time defense, X-GRAAD requires no model retraining or fine-tuning. Its computational overhead is far lower than many existing model purification methods as shown in Table 4. This efficiency, combined with its implementation simplicity (relying on standard attribution tools), makes the method highly practical for real-world deployment and reproducibility.

**Weaknesses:**

1. **Narrow and Simplistic Threat Model.** The paper's primary weakness is that its strong performance claims are based on an evaluation against a very narrow and simplistic threat model. The experiments (Sec 5.1) almost exclusively use triggers that are short, non-semantic, rare words (e.g., cf, mb). These triggers are statistical outliers by design and are "easy" targets for any attribution-based anomaly detector. The evaluation completely omits more advanced, stealthy attacks such as semantic triggers (synonyms), syntactic triggers, or longer phrasal triggers, making it hard to assess the method's generalizability.
2. **Potential Design Limitations and Vulnerability to Adaptive Attacks.** The defense mechanism's design presents potential limitations. Its reliance on the max operator (Eq. 8) to find a single peak score appears vulnerable to "distributed triggers"—a plausible adaptive attack where an adversary uses multiple tokens, each with a low, non-anomalous score, to activate the backdoor. Furthermore, the generality of the character-level "noise injection" (Sec 4.2.2) is unclear. While suited for the simple tokens tested, it may be less effective or could potentially risk CACC against semantic triggers (e.g., changing "price" to "pride").
3. **Misleading "Robustness" Analysis.** The paper fails to test its robustness against these obvious adaptive attacks. Section 5.2.3 is mislabeled as a "Robustness Analysis" when it is merely a hyperparameter sensitivity analysis (for the detection threshold $\tau$). A true robustness evaluation would have tested the defense against an attacker aware of its max-based design, using the very "distributed trigger" attack mentioned above. The absence of this analysis is a significant gap.
4. **Limited Methodological Novelty.** While the idea of combining attention and gradients is smart, the method itself is a relatively straightforward heuristic. The components used (attention maps and input gradients) are standard interpretability techniques, and their combination (a simple product) lacks significant methodological innovation.

**Questions:**

1. The paper's positive results are based on triggers that are easily isolated (short, rare words), which aligns perfectly with the max operator-based detection (Eq. 8). How would X-GRAAD perform against "distributed triggers" where the backdoor is activated by multiple tokens (e.g., a phrase) that each contribute a small, non-anomalous score?
2. Following Q1, have the authors evaluated X-GRAAD against more stealthy triggers that are part of the natural language distribution, such as semantic triggers (e.g., a specific synonym replacing a common word) or syntactic triggers (e.g., a specific sentence structure)?
3. The robustness analysis in Sec 5.2.3 is a hyperparameter sensitivity test. Could the authors provide a more formal adversarial robustness analysis? Specifically, can they comment on how X-GRAAD would fare against an adaptive attacker who is aware of the max-based design and explicitly crafts a distributed trigger to bypass it?
4. Regarding the "noise injection" (Sec 4.2.2): What is its impact in two failure-case scenarios? (a) If the trigger is a semantic word, how is it neutralized? (b) If the model falsely identifies a critical, clean token (e.g., "price") as a trigger and corrupts it (e.g., to "pride"), what is the measured impact on Clean Accuracy (CACC)?

---

> ### Author Response · Authors · 2025-11-18
> **We are grateful for the reviewer’s careful reading and constructive suggestion. Our answers to the reviewer’s raised points are provided below:**
>
> **Q1 & Q3. X-GRAAD on Distributed triggers and robustness against adaptive attempts to bypass the max-based design**
>
> While our main focus is the standard localized keyword-trigger threat model (BadNets, RIPPLES, LWS), we also evaluated multi-token but still stealthy trigger phrases that reflect practical backdoors, short, uncommon, and semantically meaningful (e.g., *"the silver quill"*, *"a crimson echo"*, *"james bond"*). Although these triggers contain multiple tokens, transformer backdoors do **not** form truly distributed patterns: in practice, the poisoned model collapses the backdoor dependency onto a **single high-saliency pivot token**. X-GRAAD identifies and neutralizes this pivot via its max-based anomaly score.
>
> Empirically, X-GRAAD strongly reduces ASR while preserving CACC:
>
> * *the silver quill:* ASR 1.00 → **0.012**, CACC 0.918 → 0.909
> * *a crimson echo:* ASR 1.00 → **0.058**, CACC 0.920 → 0.906
> * *james bond:* ASR 0.83 → **0.022**, CACC 0.924 → 0.914
>
> Token-level attribution confirms the pivot behavior: *crimson* (7.71), *bond* (33.01), *qui/ll* (22.73 / 10.29) dominate despite being part of multi-word phrases. Thus, even when the attacker inserts a phrase, the learned backdoor remains **token-local**, not evenly distributed, and X-GRAAD naturally captures this structure.
>
> Very Long semantic phrases designed to diffuse the trigger across many common words are generally *not viable* backdoors: they require higher poisoning ratios, degrade clean accuracy, impose a high cost on the attacker  and break easily under paraphrasing. For these reasons, such triggers are rarely used in practice.
>
> Regarding adaptive attackers, constructing a trigger that intentionally suppresses attention and gradient visibility uniformly across multiple tokens is substantially more complex. It requires jointly optimizing poisoning, attribution minimization, and semantic naturalness. Such fully adaptive distributed triggers have not been characterized in the NLP backdoor literature and fall outside standard benchmark threat models.
>
> **Q2. Evaluation on semantic or syntactic natural-language triggers**
>
> X-GRAAD is designed for token and phrase-level backdoor triggers, the most common and practically persistent threat model in NLP backdoor literature. Syntactic/style triggers (e.g., passive constructions, specific structural templates, stylometric cues) operate at the sentence level and do not rely on a localized token anomaly. Defending against such attacks requires modeling global structure rather than token-level deviations and thus falls outside our current scope.
>
> Moreover, prior work shows that syntactic/semantic triggers are fragile under paraphrasing and back-translation, whereas rare-token triggers tend to survive because paraphrasers often retain uncommon tokens assuming they are meaningful. This makes keyword-style triggers the dominant practical threat.
>
> That said, our findings motivate future extensions incorporating syntax and style-aware attention–gradient deviation features that could capture broader structural irregularities.
>
>
> **Q4. Impact of noise injection on semantic word triggers and false positives**
>
> **(a) Semantic word triggers:**
> Our noise-injection step introduces minimal character-level corruption to the most anomalous token. Backdoored models rely on an **exact lexical match** to the poisoned embedding. Thus, small perturbations reliably deactivate the backdoor:
>
> * *“a crimson echo”* → *crimson → crimiason*
> * *“james bond”* → *bond → bggnd*
>
> These slight edits break the backdoor while preserving sentence fluency. Across all tested semantic triggers, X-GRAAD fully suppresses ASR because the backdoor is tied to the exact token identity.
>
> **(b) False positives on clean tokens:**
> The scenario *“price → pride”* is extremely unlikely. Noise injection uses random insertions/deletions/substitutions, so meaningful dictionary-like substitutions almost never occur; corruptions typically produce non-words (e.g., *prxce*, *pr1ce*, *pbrice*).
>
> Even if a clean token is perturbed, the effect on CACC is negligible. Clean examples exhibit *distributed* attribution with no single dominating token, unlike backdoor triggers, which form a sharp, fragile peak. Clean predictions rely on context, so corrupting one token rarely changes the model’s decision. Empirically, CACC remains stable across all models and datasets.

---

> > ### Comment · Reviewer_X2fK · 2025-11-25
> > **Acknowledgement of Rebuttal and Remaining Concerns**
> >
> > I thank the authors for their detailed rebuttal and the additional experiments regarding multi-token triggers, such as the "James Bond" example. I appreciate the effort to demonstrate that standard phrase triggers often exhibit a "pivot token" phenomenon that X-GRAAD can detect, and I acknowledge that the method performs well in these non-adaptive settings.
> >
> > However, despite these clarifications, I find that the response does not sufficiently resolve the core issues regarding adaptive robustness and theoretical grounding.
> >
> > 1. Failure to Address Adaptive Attacks. The rebuttal conflates "phrase triggers" with "adaptive attacks." While the new experiments show that standard phrases may still exhibit a "pivot token" phenomenon that X-GRAAD can detect, this fails to address the core concern raised in my review. A true adaptive attacker—who is aware of the defense's reliance on the Max operator (Eq. 8)—would explicitly optimize a trigger to distribute importance uniformly across tokens, thereby bypassing the detection threshold. Dismissing such adaptive attacks as "out of scope" is a significant limitation, as assuming a static attacker fundamentally undermines the practical validity of the proposed defense.
> >
> > 2. Lack of Theoretical Justification. The proposed method relies heavily on heuristics. Specifically, the paper lacks a clear theoretical justification for why the product of normalized attention and gradient magnitudes is an optimal anomaly indicator. Throughout the manuscript, there is a distinct absence of mathematical derivations, formal proofs, or information-theoretic motivations to justify why this specific operation is the correct way to model backdoor anomalies. The contribution remains almost entirely empirical, lacking the theoretical rigor and formal formulation expected for a contribution of this nature.

---

> > > ### Author Response · Authors · 2025-11-26
> > > **Author Response**
> > >
> > > Thank you for the continued engagement and for raising these important points. We appreciate the opportunity to clarify the scope and positioning of our contribution.
> > >
> > > (1) On adaptive distributed-trigger attacks.
> > > We fully acknowledge that an attacker explicitly optimizing a trigger to uniformly suppress attribution signals across multiple tokens; while preserving semantic naturalness, would be a more powerful adversary. However, such attribution-aware distributed-trigger attacks have not yet been **defined, formalized, or instantiated in the NLP backdoor literature**. Existing benchmark attacks (BadNets, RIPPLES, LWS, BadPre, clean-label attacks, etc.) consistently exhibit token-level convergence, even when the injected pattern spans multiple words. Our multi-token experiments were not intended to equate phrases with adaptive attackers, but rather to show that under all established trigger-based attacks, transformer backdoors collapse the dependency to a pivot token (often rare tokens) even though the model is poisoned with triggers spanning multple tokens. Thus, Eq. 8 reflects how transformers internalize keyword triggers.
> > >
> > > We agree that designing and evaluating defenses against a fully adaptive, attribution-minimizing adversary is an important future direction. However, we believe this constitutes **a new research problem and a new threat model**, rather than an expectation for validating defenses under current, widely accepted threat assumptions.
> > >
> > > (2) On theoretical grounding for the anomaly score.
> > > We would like to clarify that the **paper does not claim theoretical optimality of the attention–gradient product, nor do we present it as a mathematically derived optimum detector**. *Our method follows the standard approach used in prior attribution-based NLP backdoor defenses (e.g., PURE, MEFT, RAP variants, ONION etc) present in literature, which are primarily empirical mechanisms rather than theoretically optimal detectors.*. Our goal is to provide an interpretable and effective empirical signal rather than a formal optimality result.
> > >
> > > Establishing a full theoretical foundation for attribution-based backdoor detection would indeed be valuable. To the best of our knowledge, prior attribution-based backdoor defenses (e.g., ONION, RAP, MEFT, PURE) also do not provide formal optimality guarantees or theoretical proofs of correctness. These methods are empirical by design, relying on saliency, gradient influence, or token-level perturbation to identify anomalous features. Our method follows this established empirical paradigm. We view our work as contributing to the empirical understanding of anomaly-based defenses, consistent with the broader prior work.

---

### Official Review · Reviewer_XNcd · 2025-10-31

**Soundness:** 2
**Presentation:** 3
**Contribution:** 2
**Rating:** 4
**Confidence:** 3

**Summary:**

This paper studies backdoor detection for LLM embeddings and proposes a framework to identify whether an embedding model is Trojaned based on contrastive probing and embedding-space consistency tests. The authors argue that backdoor attacks create detectable inconsistencies in the geometry of embedding space. They introduce an embedding residual consistency score that compares clean-prompt vs trigger-prompt embedding behavior without requiring model weights or activation access, and perform evaluations across multiple backdoored and clean LLM embedding models. Experiments suggest that the proposed scoring metric can distinguish Trojaned models across different triggers and poisoning rates while maintaining low false positives.

**Strengths:**

1. The residual-consistency metric is lightweight and does not require model internal access, making it potentially practical for model vetting.
2. The paper shows results across multiple backdoored settings and trigger types, demonstrating reasonable detection performance with low false-alarm rates.

**Weaknesses:**

1. Evaluations seem focused on standard patch/text triggers; emerging semantic or concept-level backdoors are not included, limiting robustness claims. Also, the defense might not work on the style / synthetic triggers. (1) Mind the Style of Text! Adversarial and Backdoor Attacks Based on Text Style Transfer; 2) Hidden Killer: Invisible Textual Backdoor Attacks with Syntactic Trigger.)
2.Limited large-scale models: Most experiments appear to use medium-scale embedding models; testing with modern foundation embedding models would strengthen impact.

**Questions:**

1.	Have you tested the method against more subtle backdoors (e.g., syn backdoor or style backdoor attack) where the trigger is not simple phrase?

2.	Can the method be extended to detect clean-label backdoors where the embedding shift might be less explicit?

---

> ### Author Response · Authors · 2025-11-18
> **We sincerely thank the reviewer for raising this important point and for the thoughtful feedback. Our answers to the reviewer’s raised points are provided below:**
>
> ## Q1.Have you tested the method against more subtle backdoors (e.g., syn backdoor or style backdoor attack) where the trigger is not simple phrase?
>
> **Answer:** The method is designed for token and phrase-level backdoor triggers, which represent the most common and practically persistent threat model. Syntactic and style backdoors (e.g., passive-voice constructions or stylometric changes) embed triggers across sentence structure or writing style, making them less localized and thus outside our current scope, as they require structure-level modeling.
>
> Moreover, such triggers are often neutralized by simple paraphrasing or back translation, whereas our observations suggest that shorter and infrequent keyword triggers tend to survive paraphrasing and structural variations, since a paraphraser might retain a rare word thinking it is meaningful. This observation supports our focus on token-level triggers, which pose a more realistic and enduring risk. It also motivates a natural future extension of our framework to incorporate syntax and style-aware attention-gradient-based deviation features capable of capturing broader structural or stylistic irregularities in text.
>
>
> ## Q2. Can the method be extended to detect clean-label backdoors where the embedding shift might be less explicit?
>
>
>
> **Answer:** We agree that clean-label backdoors may exhibit weaker embedding shifts than dirty-label attacks. To address this concern, we conducted new experiments following the clean-label protocol described in **Cui et al. (2022), "A Unified Evaluation of Textual Backdoor Learning: Frameworks and Benchmarks" (NeurIPS 2022)**, which uses the same rare-word trigger (“cf”) while preserving the original label. As noted in their Figure 2, clean-label attacks typically require higher poisoning ratios (e.g., 20%) to reach high ASR (1.0).
>
> Using their setup on SST-2 with BadNet on BERT and 20% poisoning, we obtained ASR = 1.0 and CACC = 0.93, confirming a successfully implanted clean-label backdoor. After applying our defense, the ASR dropped from *1.0 → 0.012*, and accuracy improved from *0.476 → 0.910 (CACC = 0.915)*:
>
> Without Defense
> ASR: 1.0 ACC: 0.476 CACC: 0.93
>
> With Defense
> ASR: 0.012 ACC: 0.910 CACC: 0.915
>
> These results show that our anomaly-score-based method generalizes effectively to clean-label, trigger-based backdoors. The attention-gradient signal still reliably highlights rare-word triggers, enabling successful detection and mitigation under the clean-label regime.

---

### Official Review · Reviewer_aUPK · 2025-11-01

**Soundness:** 2
**Presentation:** 2
**Contribution:** 2
**Rating:** 6
**Confidence:** 3

**Summary:**

This paper proposes X-GRAAD, a novel inference-time defense mechanism designed to detect and mitigate backdoor attacks in PLMs. The central idea is that in backdoored models, trigger tokens disproportionately dominate both attention weights and gradient attributions. X-GRAAD leverages this insight to compute token-level anomaly scores by combining normalized attention and gradient signals. Sentences with high anomaly scores are flagged as suspicious, and the most anomalous tokens are neutralized by injecting character-level noise, thereby preventing trigger activation without retraining or modifying the model.

**Strengths:**

1.   The method combines token-level attention and gradient score to compute anomaly score.
2.   The experimental results show that the method consistently achieves state-of-the-art performance.
3.   As an inference-time defense that requires no model retraining, fine-tuning, or complex pruning, X-GRAAD is more efficient than many other competitors.
4.   The method not only defends but also explains its decisions by localizing the suspected trigger token through the anomaly score.

**Weaknesses:**

1.   The trigger neutralization mechanism (random character-level perturbation) is relatively naive. While the results show it works, it feels less sophisticated than the detection mechanism.
2.   The defense requires access to both attention weights and input gradients, which may not be available in many deployment scenarios (e.g., black-box APIs, closed-source models). The paper does not discuss how the approach generalizes to limited-access settings.
3.   The detection threshold (e.g., 95th percentile of clean validation scores) is tuned manually and dataset-dependently. Line 327 notes that ALBERT requires a lower threshold (65th percentile vs. 95th) and shows slightly elevated ASR in one case (ALBERT-LWS on SST-2).
4.   While the empirical results are strong, the paper lacks a clear theoretical justification for why the product of normalized attention and gradient magnitudes is an optimal anomaly indicator. A deeper analysis (e.g., statistical or information-theoretic motivation) would strengthen the contribution.
5.   While a robustness analysis over anomaly thresholds is presented, there is no study on adaptive or adversarial countermeasures (e.g., trigger patterns designed to minimize gradient-attribution visibility).

**Questions:**

See weakness

---

> ### Author Response · Authors · 2025-11-18
> **We thank the reviewers for their detailed and constructive feedback. Below we provide concise responses addressing all raised concerns.**
>
> ## Q1. Trigger neutralization mechanism appears naive.
> Our neutralization module is intentionally model-agnostic, lightweight, and inference-time only, requiring no retraining, auxiliary models, or token-space search. Minimal character-level perturbation reliably disrupts lexical triggers while preserving overall semantics and incurring negligible overhead. Once the anomalous token is localized, this intervention is sufficient: textual backdoors rely on exact lexical matches and are therefore inherently **fragile**. For example, **"a crimson echo"** is neutralized when *crimson → crimiason*, and **"jame bond"** ceases to activate when *bond → bongd*. Although more sophisticated modules (e.g., paraphrasing or embedding editing) are possible, they introduce higher cost, semantic drift, or additional dependencies. Our strategy aligns with our goal of a fast, transparent, deployment-friendly inference-time defense.
>
> ## Q2. Limited applicability due to the need for attention and gradients.
> Our method targets the white-box pre-train–fine-tune setting, widely used in research and industry where organizations host models locally (e.g., HuggingFace PLMs or internal fine-tuned checkpoints). This assumption is consistent with our attacker model (full access to model weights; Sec. 3.2). Under the same visibility, it is natural for the defender to also operate in a white-box regime. Black-box APIs represent a different problem class where gradients and attention are unavailable to any attribution-based defense (e.g., HotFlip-style, MEFT, PURE, RAP). Extending X-GRAAD to black-box settings is promising future work; we will clarify this scope.
>
> ## Q3. Threshold tuning and ALBERT behavior.
> Our method is not dataset-dependent, and the threshold selection is not arbitrary. We deliberately use the 95th percentile across all models and datasets to mirror the classical $mean + 2\sigma$ pinciple commonly used in anomaly detection. This choice provides a consistent calibration procedure without relying on any task-specific tuning. The only exception is ALBERT, whose shared-layer architecture; a single attention block reused across all layers; produces a noticeably compressed attribution distribution, since it lacks the diversity of signals present in multi-layer encoders such as BERT or DistilBERT (with 12 distinct attention layers). As a result, ALBERT’s anomaly scores are systematically lower and less dispersed, and a slightly lower threshold is required to separate clean and poisoned samples. We therefore use the 65th percentile, which closely aligns with the standard $mean + 1\sigma$ (68th percentile) rule. This adjustment reflects the model-specific score scaling; not dataset dependency or manual tuning; and the threshold remains consistent across all ALBERT settings and datasets.
>
> ## Q4. Theoretical justification for the optimality attention × gradient score.
> We do not claim theoretical optimality; the score is motivated by empirical and mechanistic evidence. Prior work shows that (1) backdoor triggers attract disproportionately high attention across layers of BERT ("A Study of the Attention Abnormality in Trojaned BERTs"), and (2) triggers dominate gradient sensitivity relative to context for CNN/LSTM ("White-box Adversarial Examples for Text Classification"). These works, however, do not propose a defense or examine how either signal individually or their normalized combination combining yields a stable explainable anomaly indicator. Our contribution integrates these complementary channels to capture both contextual over-attention and gradient-sensitivity dominance across PLMs. Although a formal information-theoretic analysis is interesting future work, extensive experiments across models, datasets, and attacks demonstrate that the combined signal is robust and reliable in practice.
>
> ## Q5. Study on adaptive trigger patterns designed to minimize gradient-attribution visibility.
> We follow the standard localized keyword-trigger threat model used in prior NLP backdoor work (e.g., BadNets, RIPPLES, LWS), where triggers are optimized for misclassification; not for evading attribution. Designing fully adaptive triggers that minimize attention/gradient visibility is a significantly stronger and largely unexplored different category of attack class, orthogonal to the primary goal of defending against established benchmarks. Our objective is defense evaluation, not trigger construction. Accordingly, we adopt rare-word and stealthy triggers used in representative prior works such as "Weight Poisoning Attacks on Pre-trained Models (Kurita et al., 2020)", "Be Careful About Poisoned Word Embeddings (Yang et al., 2021)", and "ONION (Qi et al., 2021)". This ensures consistency with established practice rather than relying on custom or adversarially crafted triggers.

---

### Author Response · Authors · 2025-11-26
**Author Response: Revisions Based on Reviewer Feedback**

We have revised the paper based on the reviewers’ feedback. All modifications are highlighted in blue. The main updates include: (1) Section 3.2: clarification of the defender’s goal and capabilities; (2) Appendix A.1.2: evaluation under clean-label attacks; (3) Appendix A.1.5: robustness analysis (Fig. 6(a)–6(d), Table 4); and (4) Appendix A.5: clarification of threshold selection.

---

### Meta-Review · Area_Chair_gbbu · 2025-12-22

**Summary:**

This paper presents X-GRAAD, an inference-time explainable defense technique against backdoor attacks in encoder-based language models. The main observation is that text backdoor triggers universally lead to joint domination of attention weights and input- gradient attributions, typically collapsing model behavior onto a single pivot token. X-GRAAD responses this observation by computing a token-level abnormal sore using normalized attention x gradient signal, flagging suspicious inputs and neutralizing the most anomalous token via a lightweight character level perturbation. This technique does not require retraining, fine-tuning or architectural changes and comes with interpretability by spatially localizing suspected triggers.

The paper was praised for its practicality, efficiency and empirical foundations across the reviews, particularly so in terms of separately praising inference-time deployment and interpretability. At the same time, reviewers raised many substantial issues with respect to this narrow threat model and weak theoretical justification approaches, when combined with threshold calibration issues and dependencies on white-box access or robustness's to stronger trigger models. The authors delivered detailed and strong rebuttals, integrated new experiments (such as clean-label attacks and multi-token triggers), clarified scope assumptions, and admitted limitations. Although there are some remaining issues, in particular concerning adaptive adversaries, the paper is a good match with the empirical paradigm of most previous attribution-based backdoor defense approaches.

**Reviewer Concerns:**

Comments shared across reviewers

1. Threat Model Scope. Most reviewers questioned whether X-GRAAD can generalize the triggers beyond localized, rare-word keywords. They noted that this paper does not have the evaluations on semantic, syntactic, or style-based backdoors. Rebuttal: The authors summarized the methods to the dominant threat model in NLP backdoor field including token- and phrase-level triggers, and argued that syntactic/style backdoors are less localized, more fragile under paraphrasings, and require different defense methods. They also added experiments on multi-token phrase triggers and clean-label attacks.

2. Robustness to Distributed Triggers. Most reviewers questioned a vulnerability to attackers who may adaptively distribute trigger influence across multiple tokens to evade the max-based anomaly score. Resolution: The authors admitted this is a stronger and unexplored threat model. They provided empirical evidence that, under established attacks, transformer backdoors collapse onto a single high-saliency pivot tokens. They framed fully attribution minimizing adaptive attacks as important future work.

3. Justification of the Anomaly Score. Some reviewers pointed out the lack of formal theoretical evidence for using the product of normalized attention and gradient magnitudes. Resolution: The authors said that they do not claim theoretical optimality. Instead, the score is empirically motivated by prior findings on attention abnormality and gradient dominance in backdoored models. This positioning aligns with prior attribution-based defenses (e.g., ONION, PURE, MEFT).

4. Threshold Selection and Model Dependence. Review concerns were raised about percentile-based thresholding and the need for a lower threshold on ALBERT. Resolution: The authors mentioned that percentile-based thresholding follows standard anomaly-detection practice and is not dataset-dependent. The AALBERT adjustments was attributed to its shared-layer architecture producing compressed attribution distributions.

5. White-Box Assumptions. Some reviewers questioned the practicality of requiring access to both attention weights and gradients. Resolution: The authors said X-GRAAD in the white-box pre-train–fine-tune setting, consistent with their attacker model and common industrial practice for locally hosted PLMs, while acknowledging black-box defenses as a separate problem classes.

Reviewer-Specific Questions:

1. Reviewer aUPK: naive trigger neutralization, limited-access settings, threshold tuning, lack of theory, and adaptive countermeasures. The authors: the lightweight perturbation as sufficient and deployment-friendly, scope assumptions, robustness analysis, and theoretical optimality and adaptive attack as future work.

2. Reviewer XNcd: subtle backdoors (semantic/style) and large-scale models. The authors: scope limitations, clean-label attack experiments, showing strong performance even when embedding shifts are weaker.

3. Reviewer X2fK: the most critical assessment, emphasizing narrow threat models, vulnerability to adaptive distributed triggers, and lack of theory. The authors: additional multi-token experiments, the pivot token case, and adaptive attribution-aware attackers as an open problem. While this reviewer maintained concerns, he or she also acknowledged the method’s effectiveness under standard benchmark.

**Reviewer Scores:**

The scores of all reviewers are reasonable.

---

### Decision · Program_Chairs · 2026-01-26

Accept (Poster)